# Stage-Specific L-Proline Uptake by Amino Acid Transporter Slc6a19/B^0^AT1 Is Required for Optimal Preimplantation Embryo Development in Mice

**DOI:** 10.3390/cells12010018

**Published:** 2022-12-21

**Authors:** Tamara Treleaven, Matthew Zada, Rajini Nagarajah, Charles G. Bailey, John E. J. Rasko, Michael B. Morris, Margot L. Day

**Affiliations:** 1School of Medical Sciences, Faculty of Medicine and Health, The University of Sydney, Camperdown, NSW 2006, Australia; 2Gene & Stem Cell Therapy Program Centenary Institute, The University of Sydney, Camperdown, NSW 2050, Australia; 3Faculty of Medicine and Health, The University of Sydney, Camperdown, NSW 2006, Australia; 4Cancer & Gene Regulation Laboratory Centenary Institute, The University of Sydney, Camperdown, NSW 2050, Australia; 5Cell & Molecular Therapies, Royal Prince Alfred Hospital, Missenden Rd, Camperdown, NSW 2050, Australia

**Keywords:** proline, amino acid transporter, *Slc6a19*, B^0^AT1, preimplantation embryo, mouse

## Abstract

L-proline (Pro) has previously been shown to support normal development of mouse embryos. Recently we have shown that Pro improves subsequent embryo development when added to fertilisation medium during in vitro fertilisation of mouse oocytes. The mechanisms by which Pro improves embryo development are still being elucidated but likely involve signalling pathways that have been observed in Pro-mediated differentiation of mouse embryonic stem cells. In this study, we show that B^0^AT1, a neutral amino acid transporter that accepts Pro, is expressed in mouse preimplantation embryos, along with the accessory protein ACE2. B^0^AT1 knockout (*Slc6a19^−/−^*) mice have decreased fertility, in terms of litter size and preimplantation embryo development in vitro. In embryos from wild-type (WT) mice, excess unlabelled Pro inhibited radiolabelled Pro uptake in oocytes and 4–8-cell stage embryos. Radiolabelled Pro uptake was reduced in 4–8-cell stage embryos, but not in oocytes, from *Slc6a19^−/−^* mice compared to those from WT mice. Other B^0^AT1 substrates, such as alanine and leucine, reduced uptake of Pro in WT but not in B^0^AT1 knockout embryos. Addition of Pro to culture medium improved embryo development. In WT embryos, Pro increased development to the cavitation stage (on day 4); whereas in B^0^AT1 knockout embryos Pro improved development to the 5–8-cell (day 3) and blastocyst stages (day 6) but not at cavitation (day 4), suggesting B^0^AT1 is the main contributor to Pro uptake on day 4 of development. Our results highlight transporter redundancy in the preimplantation embryo.

## 1. Introduction

Amino acids (AAs) play important roles in mammalian preimplantation development [1,2,3,4]. Reproductive fluid in the oviduct and uterus contains a mixture of all essential (EAA) and non-essential AAs (NEAA) which support embryo development from fertilisation to implantation [5,6]. During in vitro culture of mouse embryos, addition of NEAAs to the medium promotes cleavage to the 8-cell stage [1,7] whereas addition of EAAs has no effect until after the 8-cell stage, when they promote an increased number of cells in the inner cell mass (ICM) [1].

Some AAs, such as alanine, glycine, glutamine and taurine, are present at millimolar concentrations in reproductive tract fluid [5] and can be accumulated to the same or higher concentrations within the developing embryo to prevent cell volume changes caused by the high osmolality (300–310 mOsm/kg) of reproductive fluid [8]. In addition to their role as organic osmolytes, AAs are also important for protein synthesis [9], as an energy source for production of ATP [10], for buffering of intracellular pH, and regulation of reactive oxygen species (ROS) either by use in the production of glutathione or by direct ROS scavenging [11]. L-proline (Pro), for example, is able to directly scavenge ROS and when added to culture medium (in the absence of all other AAs), at a concentration similar to that found in reproductive fluid [6], improves mouse preimplantation embryo development [4]. Recently, we have shown that adding Pro and/or its analogue pipecolic acid to fertilisation medium during in vitro fertilisation (IVF) of oocytes increases subsequent embryo development, blastocyst formation and the number of cells in the ICM [12]. The improvement in development induced by Pro may be attributed to a reduction of mitochondrial activity and ROS levels in oocytes [12]. Collectively, these and other results show that Pro (and its metabolites) act in a growth factor-like manner by a variety of interconnected mechanisms including signal transduction activation, regulation of ROS levels and mitochondrial activity, and modulation of the epigenetic landscape as shown either in embryos themselves [4,12] or using embryonic stem (ES) cells as an in vitro model of embryo development [13,14,15,16].

For AAs to have a beneficial effect on preimplantation development they must be taken up into the embryo via AA transporters (AATs). During early embryo development, AATs are expressed in a stage-specific manner reflecting, in part, the changing nutritional needs of the developing embryo. For example, there are multiple transporters for Pro in the preimplantation embryo, which are expressed at various stages. In the cumulus oocyte complexes (COCs), Pro can be taken up by cumulus cells surrounding the oocyte [17] via y+LAT2 (*Slc7a6*) [18] and GlyT (*Slc6a5* and/or *Slc6a9*) [19] and Pro is then transferred to the oocyte via transzonal gap junctions. In oocytes themselves, Pro is taken up by one or more transporters, including GlyT1 (*Slc6a9*) [19,20], PROT (*Slc6a7*), and PAT1 and/or 2 (*Slc36a1*, *Slc36a2*) [12]. After fertilisation, Pro is transported via SIT1 (*Slc6a20*), which is active in the zygote and 2-cell stage [21], as well as by GlyT1, which is active in 8-cell and blastocyst stages [20]. System A transporters are the major transport system in the ICM [22]. In mouse ES cells, Pro uptake by SNAT2 (*Slc38a2*) promotes differentiation [13,23].

Improved development of zygotes to the blastocyst stage induced by Pro in culture medium is prevented by the addition of a molar excess of glycine (Gly), betaine (Bet) and leucine (Leu) [4]. This profile for AA transport is consistent with that for the transporter B^0^AT1 (*Slc6a19*). In somatic cells, B^0^AT1 is expressed in the kidney, intestine, colon and prostate, and is a major transporter of neutral AAs, including Pro, across the plasma membrane [24]. Autosomal recessive inheritance of *SLC6A19* mutations in humans leads to poor absorption of AAs from the gut, or poor retention of AAs by the kidneys, causing Hartnup disorder [24,25]. Mice in which *Slc6a19* has been knocked out (*Slc6a19*^−/−^) [26] exhibit symptoms of neutral aminoaciduria characteristic of Hartnup disorder. However, the effect of the loss of B^0^AT1 expression on reproductive capacity and in vitro development of embryos lacking B^0^AT1 has not been reported. 

In this study, we therefore investigated whether B^0^AT1 is expressed during mouse preimplantation embryo development, along with known accessory proteins, and examined the role of B^0^AT1 in fertility and embryo development by utilising B^0^AT1 knockout (*Slc6a19*^−/−^) mice. Our findings show that B^0^AT1 is expressed throughout preimplantation development. The accessory protein ACE2 was also expressed, while its paralogue collectrin (TMEM27) was not present at any pre-implantation stage. *Slc6a19*^−/−^ mice had decreased litter size compared to wild-type (WT). In addition, the development of embryos obtained from *Slc6a19*^−/−^ mice to the blastocyst stage was decreased compared to embryos from WT mice, suggesting an important role for B^0^AT1 in normal embryo development.

## 2. Methods

### 2.1. Animals (Mus musculus) 

Generation N8 *Slc6a19*^−/−^ mice [26] were fully backcrossed to N10 on C57Bl/6 background (Centenary Institute). Wild-type C57Bl/6 mice were obtained from Australian BioResources. Outbred Quackenbush Swiss (QS) mice (Animal Resource Centre, Perth, WA, Australia and Lab Animal Services, The University of Sydney, NSW, Australia) were housed under a 12 h light: 12 h dark cycle and experiments were performed under the Australian Code of Practice for the Care and Use of Animals for Scientific Purposes and in accordance with the University of Sydney Animal Ethics Committee, as required by the NSW Animal Research Act (5583 and 824) and by Sydney Local Health District Animal Ethics Committee (2016-031). 

### 2.2. Isolation of Oocytes and Embryos

Female mice (4–8 weeks of age) were superovulated by intraperitoneal injection of 10 IU pregnant mare serum gonadotropin (PMSG; Intervet, Vic., Australia) followed by 10 IU human chronic gonadotropin (hCG; Intervet) 48 ± 2 h later. Immediately after hCG injection, female mice were individually housed and paired with QS, C57BL6 (WT) or *Slc6a19*^−/−^ male mice (2–8 months of age). *Slc6a19*^−/−^ embryos were obtained by mating *Slc6a19*^−/−^ female with *Slc6a19*^−/−^ male mice. Female mice were sacrificed by cervical dislocation at either 13, 24, 46, 56, 68, 80 or 92 h post-hCG to obtain oocytes, zygotes, 2–, 4–, 5–8-cell or morula stage embryos or blastocysts, respectively. Unfertilised oocytes and embryos were dissected from the oviduct or flushed from the reproductive tract into HEPES-buffered modified human tubal fluid (HEPES-modHTF). Bovine serum albumin (BSA; 0.3 mg/mL) was added to all media at a reduced amount to minimise the possibility of BSA breakdown into free AAs in the media. The concentration of NaCl in media was adjusted to give an osmolality of 270 mOsm/kg, as described previously [4]. Cumulus cells were removed by gently pipetting oocytes with a fine glass pipette for 1 min in HEPES-modHTF + 1 mg/mL hyaluronidase, followed by washing oocytes through three drops of modHTF + 0.3 mg/mL BSA. Denuded oocytes and embryos were allocated for embryo culture, L-[^3^H]-Pro uptake or immunostaining experiments.

### 2.3. Culture of Zygotes from WT and Slc6a19^−/−^ Mice In Vitro 

Freshly isolated zygotes (day 1) were washed through at least three drops of HEPES-modHTF in the absence of AAs and then cultured in 96-well plates (Corning, NY, USA) to the blastocyst stage (120 h, day 6) at low density (1 embryo/100 μL) in modHTF + 0.3 mg/mL BSA ± 0.4 mM Pro at 37 °C in humidified air with 5% CO_2_ under mineral oil. Embryos were scored for developmental stage every 24 h. The concentration of Pro used was previously shown to improve in vitro embryo development [4] and is similar to its physiological concentration in fluid of the reproductive tract [6]. Culture experiments were repeated 4–6 times with 5–21 embryos per treatment group per experiment.

### 2.4. Measurement of L-[^3^H]-Pro Uptake in Oocytes and Embryos 

To measure the uptake of L-[^3^H]-Pro from medium, groups of 1–4 freshly isolated oocytes or embryos were allocated to 20 μL drops containing 1 μM L-[^3^H]-Pro (L-[2,3,4,5-^3^H]-proline (1 mCi/mL; Perkin-Elmer, Vic., Australia, NET483001MC) in HEPES-modHTF ± molar excess unlabelled AAs (5 mM for all AAs except Pro, which was used at 0.4 mM) covered with mineral oil and incubated for 60 min at 37 °C in humidified air with 5% CO_2_. The L-isomer was used for all unlabelled chiral AAs (Sigma Aldrich, St. Louis, MO, USA). A time-course experiment was previously performed to show L-[^3^H]-Pro uptake was linear for at least 60 min (data not shown; [12]). After incubation in each treatment, oocytes and embryos were immediately washed through cold HEPES-modHTF media (4 °C) and aspirated onto a filter mat (Perkin-Elmer, Vic., Australia, #1450-421) with 4 mL scintillation fluid (Perkin-Elmer, Vic., Australia, #6013371). Each sample group was analysed for 30 min in a MicroBeta TriLux Plate Counter (PerkinElmer, Vic., Australia). A standard curve was created from a serial dilution of 1 µM L-[^3^H]-Pro and fitted by linear regression to enable calculation of uptake in fmol min^–1^ oocyte^–1^ (or embryo^–1^) for each separate experiment.

### 2.5. Immunoflurescent Staining and Confocal Microscopy of COCs and Embryos 

After isolation or following culture, oocytes and embryos were fixed in 4% (*w*/*v*) paraformaldehyde (PFA) for 30 min at room temperature then washed three times in PBS + 1 mg/mL polyvinyl acid (PBS + PVA; Sigma-Aldrich, St. Louis, MO, USA). Oocytes/embryos were permeabilised with PBS + PVA + 0.3% (*v*/*v*) Triton X-100 for 30 min and blocked by incubation in PBS + PVA + 0.1% (*v*/*v*) Tween-20 + 0.7% (*w*/*v*) BSA (PPTB) for 30 min at room temperature. Primary antibodies were diluted 1 in 200 in PPTB and included polyclonal chicken anti-human B^0^AT1 (custom antibody from Aves Laboratories; immunising peptide CZ-DPNYEEFPKSQK representing aa 564–574 of B^0^AT1 protein [27]), rabbit anti-human ACE2 (Abcam, Cambridge, UK, Cat. No. ab15348; RRID:AB_301861) and mouse anti-collectrin (Alexis Biochemicals, San Diego, CA, USA). Appropriate control sera for each staining included pre-immune IgY serum (chicken) and purified IgG (rabbit and mouse). Oocytes and embryos were incubated in primary antibody for 2 h at room temperature and washed three times in PPTB. Secondary antibodies used were Alexa Fluor 594-coupled goat anti-chicken IgG (diluted 1:500), Alexa Fluor 488-coupled goat anti-rabbit IgG (diluted 1:200) and anti-mouse IgG (diluted 1:200). For staining of embryos for only B^0^AT1, the secondary antibody was diluted in PPTB containing FITC-conjugated phalloidin (1:200, Invitrogen, Carlsbad, CA, USA, Cat. No. F432). Oocytes and embryos were incubated in the dark for 1 h at room temperature and washed three times in PPTB then mounted in 3 μL Vectashield containing 1.5 μg/mL DAPI (Vector Laboratories, Newark, CA, USA). Images of oocytes and embryos were taken using confocal microscopy (LSM 510 Meta, Carl Zeiss, Oberkochen, Germany) using 405, 488 and 516 nm lasers and a 40× objective. Images were prepared using Fiji by Image J and are representative of 6–9 embryos stained from three separate embryo isolations.

### 2.6. Statistical Analyses 

Embryo culture experiments were performed 4–6 times with 5–21 embryos per treatment group per experiment. The number of embryos developing to a particular stage was summed together from each replicate and the percentage development calculated. The total number of embryos in each group is provided in each figure and differences between treatment groups were determined by chi-square analysis. AA uptake experiments were conducted at least three times with 1–4 oocytes or embryos per treatment group. The average uptake of L-[^3^H]-Pro for each treatment was calculated and the difference between groups compared by Student’s *t*-test or 1-way ANOVA with Dunnett’s post hoc test, as indicated in the figure legends. Data analysis was performed using GraphPad Prism v7.

## 3. Results

### 3.1. B^0^AT1 Is Expressed in Preimplantation Embryos

The stage-specific expression of the B^0^AT1 transporter in the preimplantation embryo was investigated by immunostaining. Embryos were counterstained with phalloidin, which binds to F-actin including subcortical F-actin, thereby enabling the identification of the periphery of each blastomere. Expression of B^0^AT1 was detected with F-actin at or near the plasma-membrane of embryos at all developmental stages examined, but not at the surface of contact between blastomeres (Figure 1). In 1-cell through to 8-cell stages, B^0^AT1 staining was observed in the cytoplasm close to the plasma membrane. B^0^AT1 was also detected in the apical and basolateral membranes of trophoblast cells and at the plasma membrane of inner cell mass cells in blastocysts (Figure 1).

### 3.2. The B^0^AT1 Accessory Protein ACE2 Is Expressed in the Preimplantation Embryo

Trafficking of B^0^AT1 to the plasma membrane is dependent on the co-expression of an accessory protein, either angiotensin converting enzyme 2 (ACE2) or collectrin [28,29,30]. Embryos were therefore immunostained with the B^0^AT1 antibody together with either ACE2 or collectrin antibodies to investigate whether one or both accessory proteins associated with B^0^AT1 in preimplantation embryos. B^0^AT1 colocalised with ACE2 in all preimplantation stages up to and including the blastocyst (Figure 2A). No collectrin staining was observed at any embryonic stage (Figure 2B).

### 3.3. Fertility of Slc6a19^−/−^ Mice Is Reduced 

To confirm the role of B^0^AT1 in early embryonic development, we examined a *Slc6a19*-nullizygous mouse model previously reported by us [26]. Firstly, the absence of B^0^AT1 protein in COCs and blastocysts from *Slc6a19*^−/−^ mice was confirmed by immunostaining (Figure 3A)**.** In viable mice, litter size in *Slc6a19*^−/−^ mice (KO × KO; 3.8 ± 1.5 (mean ± SD)) was significantly reduced compared to that in both WT (WT × WT; 6.4 ± 1.3 (mean ± SD), *p* < 0.001) and heterozygous mice (het × het; 6.2 ± 2.2 (mean ± SD), *p* < 0.0001) (Figure 3B), indicating B^0^AT1 deficiency negatively impacts fertility and/or live birth rate. 

### 3.4. Pro Uptake by Embryos from WT and Slc6a19^−/−^ Mice Is Dependent on Developmental Stage

The rate of uptake of L-[^3^H]-Pro by oocytes and 4–8-cell stage embryos from WT and *Slc6a19*^−/−^ mice was examined in the absence (No AA) and presence of excess unlabelled Pro. In both of these developmental stages from WT mice, excess unlabelled Pro significantly decreased the rate of L-[^3^H]-Pro uptake (Figure 4), indicating the presence of one or more AAT(s) for Pro. Whereas in oocytes and 4–8 cell embryos from *Slc6a19*^−/−^ mice, unlabelled Pro failed to prevent L-[^3^H]-Pro uptake (Figure 4), suggesting B^0^AT1 is normally an active AAT at these stages in WT mice. Any uptake now in the *Slc6a19*^−/−^ mice is via a compensatory mechanism, which is not inhibited by the presence of excess unlabelled Pro (see Discussion).

### 3.5. Pro Is Taken up by B^0^AT1 and/or an Unknown Betaine-Sensitive Pro Transporter

B^0^AT1 is a sodium-dependent neutral AAT which accepts Pro, Leu and Ala but not Bet [32]. Previously, we have shown that PROT and PAT1/2 transporters are responsible for the uptake of Pro in oocytes [12]. Here, we also investigated the competitive effect of B^0^AT1 substrates on the rate of Pro uptake at the 4–8-cell stage. All 3 AAs significantly decreased the rate of L-[^3^H]-Pro uptake in embryos from WT mice (Figure 5A) but not in embryos from *Slc6a19*^−/−^ mice (Figure 5B). 

### 3.6. B^0^AT1 Is Needed for Pro Uptake and Optimal Development at Day 4 of Development

The effect of loss of B^0^AT1 on embryo development was examined by culturing embryos from WT and *Slc6a19*^−/−^ mice from the zygote to blastocyst stage in medium ± 0.4 mM Pro (Figure 6). On day 2 (i.e., at the 2-cell stage), there was no significant difference in development between WT and *Slc6a19*^−/−^ embryos with no AAs. Fewer *Slc6a19*^−/−^ embryos developed in medium containing no AA to the ≥5-cell stage by the appropriate time (i.e., by day 3) and to the blastocyst stage (days 5 and 6) (Figure 6A). Addition of 0.4 mM Pro to the culture medium largely rescued the decreased development of embryos from *Slc6a19*^−/−^ mice, except at day 4, when fewer embryos had cavitated, suggesting slower development at this point (Figure 6B). Note, that in embryos from WT mice, Pro significantly increased development to the appropriate stage on day 4 (*p* < 0.001). Whereas in *Slc6a19*^−/−^ embryos, Pro increased development on days 3 and 6 (*p* < 0.05) compared to embryos cultured in medium containing no AA. These data suggest that in the absence of B^0^AT1, Pro can be taken up by other transporter(s) on days 3 and 6 but not on day 4 of preimplantation development.

## 4. Discussion

In this study, we investigated the effect of Pro on embryo development to determine: (1) if Pro is taken up by preimplantation embryos; and (2) if Pro transporters are expressed in embryos from WT and *Slc6a19*^−/−^ mice. In vitro, the developmental potential of an oocyte and/or embryo is improved by the exogenous addition of Pro to the culture medium in a time-dependent manner. For example, when Pro is added to fertilisation medium, cleavage rates are not affected but improvement in development occurs at compaction through to the blastocyst stage, and blastocysts produced have more cells in their inner cell mass [12]. If Pro is added to culture medium from the zygote stage onwards, improved embryo development occurs from the 5-cell stage and the proportion proceeding to blastocyst development and hatching is increased [4]. Stage-specific uptake of Pro by AATs is needed for these Pro-mediated improvements in development. Since Gly and Leu both inhibit Pro-mediated improvement in development in culture [4], and are common substrates of the neutral AAT system B^0^, we hypothesised that B^0^AT1 (*Slc6a19*) is expressed in preimplantation embryos where it is responsible for Pro uptake. 

B^0^AT1 was detected in the plasma membrane of embryos at all stages from WT mice. B^0^AT1 was also present in the cumulus cells of COCs and suggests a role for B^0^AT1 in the increased Pro uptake into oocytes by coupled cumulus cells [19]. B^0^AT1 staining was co-localised with ACE2 at all embryo stages, suggesting a role for ACE2, rather than collectrin, in the trafficking of B^0^AT1 to the plasma membrane in embryos. 

The functional role of B^0^AT1 in uptake of Pro into embryos was determined by measuring the rate of radiolabelled Pro uptake in oocytes and 4–8-cell stage embryos and competition for Pro uptake by other substrates of B^0^AT1, including Leu and Ala [32,33]. Due to the decreased fertility of *Slc6a19*^−/−^ mice, we were unable to elucidate the complete inhibition profile of Pro transport in embryos from *Slc6a19*^−/−^ mice. AAT-mediated uptake of Pro was present in oocytes from WT mice. However, despite the fact that the rate of Pro uptake into oocytes lacking B^0^AT1 is the same as for WT, it was not inhibited by excess Pro, indicating Pro uptake in the knockout is not transporter-mediated. Other studies have shown that Pro uptake in WT oocytes can occur via a number of AATs including PROT (Slc6a7), GlyT1 (Slc6a9), PAT1 (Slc36a1) and/or PAT2 (Slc36a2) [12,19], but none of these seem to be active in this knockout since excess unlabeled Pro did not reduce uptake of labelled Pro. Thus, in this study, Pro was taken up in B^0^AT1 knockout oocytes and embryos via a non-saturable route. This may be due to transport via channels that also conduct AAs, such as the volume-sensitive anion channel known to be present in oocytes [34,35,36].

At the 4–8-cell stage, Pro was taken up by embryos from both WT and *Slc6a19*^−/−^ mice with the uptake reduced in *Slc6a19*^−/−^ mice. The AAs Leu and Ala inhibited the uptake of Pro in WT but not in embryos from *Slc6a19*^−/−^ mice. Bet, which is not a substrate for B^0^AT1 [32], also inhibited Pro uptake in 4–8-cell WT embryos. PAT1 and PAT2 transport Pro, Bet and Ala [37] and are therefore good candidates for the Bet-sensitive portion of Pro uptake in 4–8 cell embryos. Recently, we have shown that PAT1 and 2 are present in the plasma membrane of mouse oocytes and may be responsible for Na^+^-independent uptake of Pro in oocytes [12]. System A and BGT1 (*Slc6a12*), which also transport Bet, are not present in preimplantation embryos [20,38]. Further studies are needed to examine the expression profile of PAT1/2 in later embryo stages. A proportion of Pro uptake at the 4–8-cell stage could be via GlyT1, which is expressed in 4- and 8-cell embryos [39,40]. Again, none of these AATs seem to be active in the knockout. It is not clear why the Bet-sensitive uptake was also lost when B^0^AT1 was knocked out. Further investigation is needed to determine whether the AAT profile in 4–8-cell embryos represents more than one transport system. 

Development of embryos from *Slc6a19*^−/−^ mice was decreased at the ≥5-cell and blastocyst stages when the medium contained no AA. These data suggest that B^0^AT1 is required in vivo for uptake and storage of Pro in the oocyte and during fertilisation, which subsequently improves later embryo development [12]. Exogenous addition of Pro to culture improved development of embryos from *Slc6a19*^−/−^ mice at the ≥5-cell and blastocyst stages but not on day 4 when cavitation occurs. In embryos from WT mice, Pro improved development at the cavitation stage. These findings indicate that B^0^AT1 is normally expressed on day 4 of in vitro embryo development, or at a cleavage stage just before cavitation, to accumulate intracellular Pro for activation of Pro-mediated intracellular mechanisms, such as regulation of reactive oxygen species and mitochondrial activity [12,41]. Furthermore, the expression of AATs during pre-implantation development is very dynamic. We speculate that during or soon after day 4 in *Slc6a19*^−/−^ mice, new Pro transporters are now expressed, and Pro is then taken up in sufficient quantity for cavitation and further development to proceed.

There was no decrease in litter size in Het x Het intercrosses and the appropriate Mendelian distribution of litter genotypes was observed [26], suggesting the maternal reproductive tract environment of *Slc6a19^+/−^* mice enables survival of *Slc6a19*^−/−^ embryos, as seen with improved development in vitro of *Slc6a19*^−/−^ embryos in the presence of Pro. Note though, a maternal diet low in vitamin B3 in pregnant *Slc6a19^+/−^* mice results in embryo loss and malformations [42]. We postulate that both intrinsic (i.e., *Slc6a19* KO embryo) and extrinsic factors (i.e., *Slc6a19* KO reproductive tract) are responsible for the decreased litter size in KO × KO intercrosses. This has implications for pregnancy in humans who carry loss-of-function mutant alleles in *SLC6A19*. These findings suggest that embryonic development and fertility could be improved in *Slc6a19*^−/−^ mice by supplementation of the maternal diet with Pro.

This is the first study to demonstrate expression of the neutral AAT B^0^AT1 in preimplantation embryos. Absence of this transporter adversely impacts fertility and embryo development, although the effect on embryo development can be overcome by the presence of Pro in embryo culture medium. This study demonstrates the importance of Pro for embryo development and that Pro can be taken up at each developmental stage by one or more transporters. The presence of multiple AATs for Pro and the ability to compensate for the loss of an individual AAT by upregulation of non-saturable transport mechanism indicates the critical importance of Pro for embryo development.

## Figures and Tables

**Figure 1 cells-12-00018-f001:**
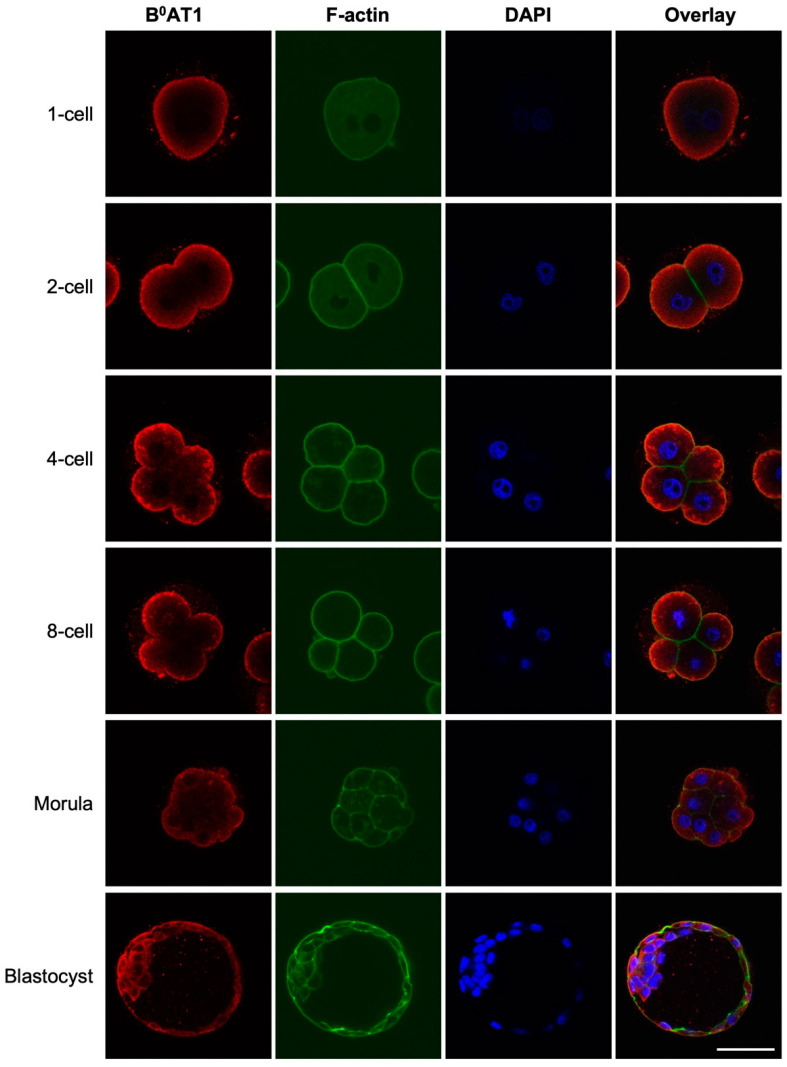
B^0^AT1 is expressed throughout preimplantation mouse embryo development. Embryos from QS mice were freshly isolated at 1-, 2-, 4-, and 8-cell, morula and blastocyst stages, fixed and immunostained for B^0^AT1 (red) and counterstained for F-actin with phalloidin (green) and nuclei with DAPI (blue). Fluorescent images were captured by confocal microscopy using a 40× objective. No staining was observed when a pre-immune IgY equivalent was used as a negative control for B^0^AT1 staining (not shown). Scale bar = 50 µm for all images.

**Figure 2 cells-12-00018-f002:**
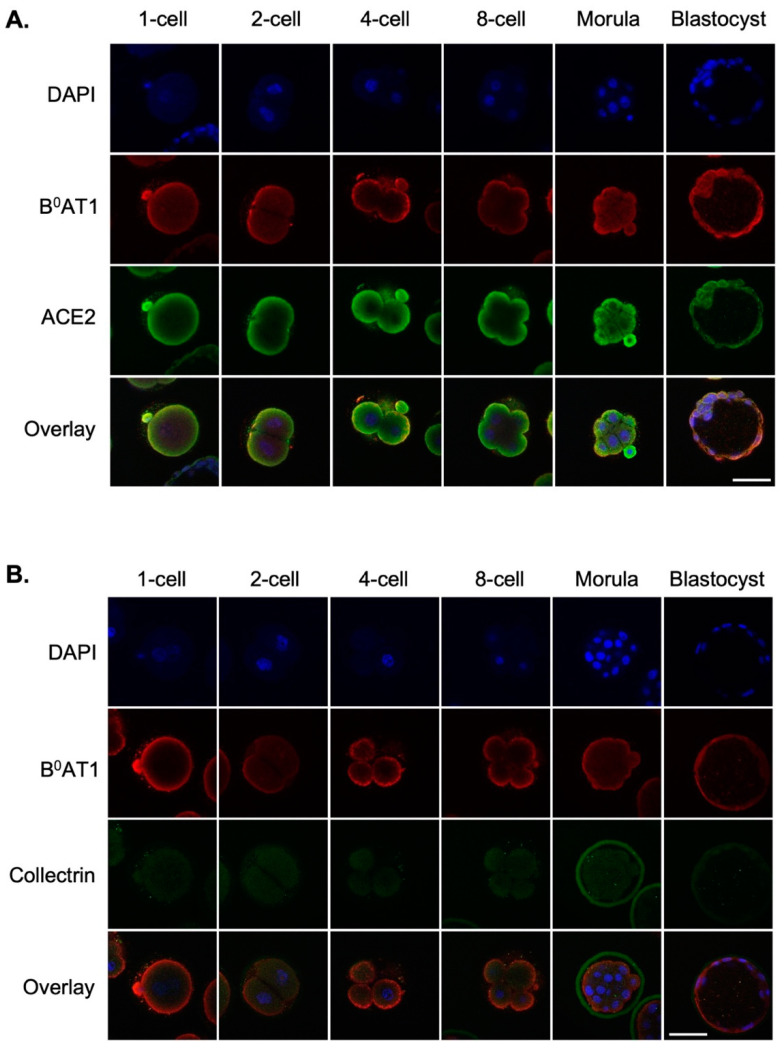
ACE2 colocalises with B^0^AT1 in the mouse preimplantation embryo. Embryos were freshly isolated from QS mice at 1-, 2-, 4-, and 8-cell, morula and blastocyst stages, fixed and immunostained for B^0^AT1 (red) and (**A**) ACE2 (green), or (**B**) collectrin (green)**.** Nuclei were counterstained with DAPI (blue). Fluorescent images were captured by confocal microscopy using a 40× objective. No staining was observed when a pre-immune IgY or IgG equivalent was used as a negative control for the primary antibodies (not shown). Scale bar = 50 µm for all images.

**Figure 3 cells-12-00018-f003:**
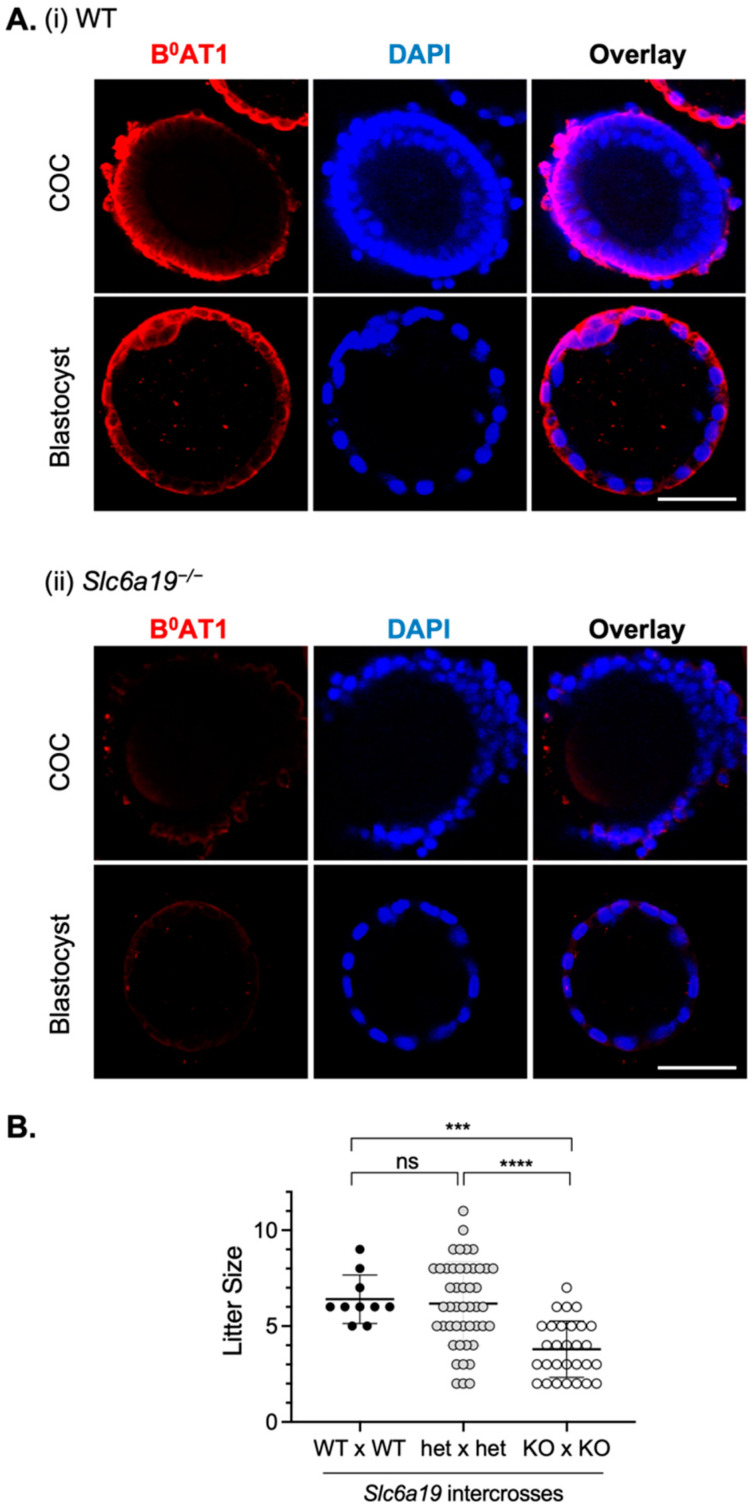
B^0^AT1 deficiency impairs mouse fertility. (**A**) COCs and blastocysts were freshly isolated from (**i**) WT and (**ii**) *Slc6a19*^−/−^ mice and immunostained for B^0^AT1 (red) and DAPI (blue) to confirm loss of B^0^AT1 protein in knockouts. Scale bar = 50 µm for all images. No staining was observed when a pre-immune IgY was used as a negative control for B^0^AT1 staining (not shown). (**B**) The number of pups born from wild-type (WT × WT) [31], heterozygous (het × het) and homozygous *Slc6a19*^−/−^ mice (KO × KO) intercrosses. Data show mean ± SD; *** *p* < 0.001, **** *p* < 0.0001, ns is not significant. Statistical analysis was performed using a one-way ANOVA with Tukey’s post hoc test for multiple comparisons.

**Figure 4 cells-12-00018-f004:**
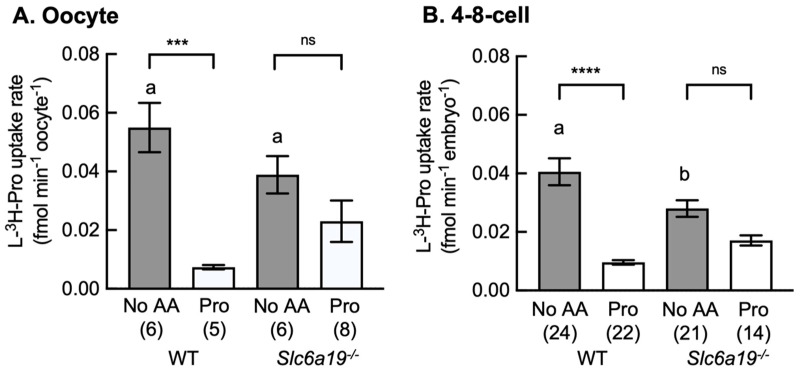
Excess unlabelled Pro reduces the rate of L-[^3^H]-Pro uptake in WT but not *Slc6a19*^−/−^ embryos. Rate of uptake of 1 µM L-[^3^H]-Pro in (**A**) oocytes, and (**B**) 4–8-cell embryos from wild-type (WT) and *Slc6a19*^−/−^ mice without (No AA) or with molar excess of unlabelled Pro (0.4 mM). Each bar represents the mean ± SEM with the number of replicates given in parentheses. Each replicate was performed on 1–4 oocytes or embryos over at least 3 independent experiments. Data were analysed by Student’s *t*-test. ‘No AA’ bars with different letters (a,b) are significantly different (*p* < 0.05). *** and **** indicate *p* < 0.001 and *p* < 0.0001 respectively; ns, not significant (*p* > 0.05) for No AA vs Pro.

**Figure 5 cells-12-00018-f005:**
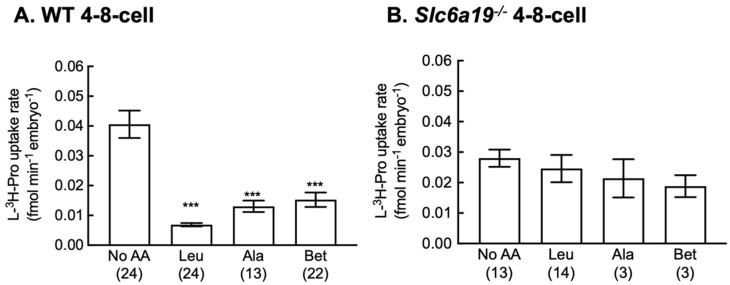
Characteristics of AA transport are different in embryos obtained from *Slc16a19^−/^^−^* mice compared to embryos from WT mice. Rate of uptake of L-[^3^H]-Pro was measured in 4–8-cell stage embryos from (**A**) WT or (**B**) *Slc6a19*^−/−^ mice without (No AA) or with molar excess (5 mM) of unlabelled AAs. Each bar represents the mean ± SEM with the number of replicates given in parentheses. Each replicate was performed on 1–4 embryos over at least 3 independent experiments. Data were analysed using one-way ANOVA with Dunnett’s post hoc test. *** *p* < 0.001 compared to No AA. Bet = betaine, Leu = leucine, Ala = alanine. Note that bars for No AA and Pro in this figure are the same data as in Figure 4.

**Figure 6 cells-12-00018-f006:**
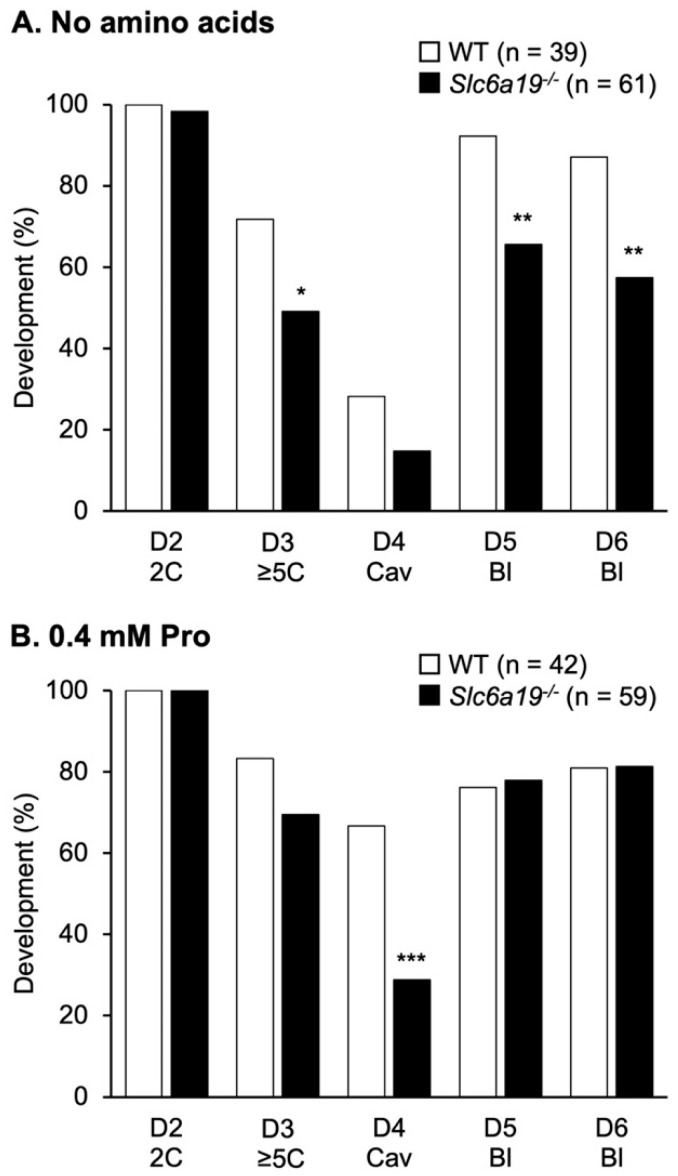
Pro improves development of *Slc6a19*^−/−^ embryos at ≥5–8-cell and blastocyst stages. Zygotes were cultured in the (**A**) absence of AAs, or (**B**) in the presence of 0.4 mM Pro and scored for development to the appropriate stage every 24 h (i.e., to the 2-cell (2C) on day 2 (D2), 5–8-cell (≥5C) on D3, cavitation (Cav) on D4, and blastocyst stage (Bl) on D5 and D6. Data on each day of development were compared by chi-square analysis of pooled data from 4–6 replicate experiments, with the total number of embryos in each group given in parentheses. Bars with an asterix are significantly different to WT at each timepoint: * *p* < 0.05, ** *p* < 0.01 and *** *p* < 0.001.

## Data Availability

All data are included in the published article. Data and materials will be made available upon request.

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
