# Peer review of "Stage-Specific L-Proline Uptake by Amino Acid Transporter Slc6a19/B0AT1 Is Required for Optimal Preimplantation Embryo Development in Mice"

_cells, 2022, doi:10.3390/cells12010018_

Round 1

Reviewer 1 Report

The manuscript titled “Stage-specific L-proline uptake by amino acid transporter B0AT1 is required for optimal preimplantation embryo development in mice ” by Treleaven et al. examined how proline added to the culture medium, mimicking the fluid in the oviduct, is taken up by the embryo and how much the added proline effect on embryonic development.

They showed that among the amino acid transporters capable of proline uptake, B0AT1, a neutral amino acid transporter, is expressed in preimplantation embryos; the knockout mice of Slc6a19, which encodes B0AT1, have reduced litter size; the uptake of amino acids, including proline, is altered in Slc6a19 KO embryos; and that the addition of proline to culture medium improves the preimplantation development of Slc6a19 KO embryos.

 Overall, the paper is well written and the statistical analysis of data and presentation of figures is appropriate. The following questions should be answered before publication.

 Q1At line 231 on page 7, “In viable mice, litter size in Slc6a19-/- mice (KO x KO; 3.8 +/- 1.5) was significantly reduced compared to that in heterozygous mice (het x het; 6.2 +/- 2.2 (mean SD), P < 0.0001) (Figure 3B) indicating B0AT1 deficiency negatively impacts fertility/and or live birth rate.”

The authors only analyzed KO x KO and het x het, but theoretically 25% of embryos in het x het should be KO embryos. Given that KO embryos show developmental defects, het x het does not appear to be a good control. Is the number of litters in het x het lower than that in WT x het, het x WT and WTx WT?

Q2. In vitro culture systems have also shown reduced embryonic development with Slc6a19 KO (Fig. 6), but this is no AA condition and does not reflect natural oviductal conditions. Does the efficiency of blastocyst formation change with Slc6a19 KO in natural mating? Would that explain the reduced litter size of KO x KO (or potentially het x het) embryos?

Q3. Comparing proline uptake (no AA condition) in wild-type and KO embryos in Fig. 4, there is no difference in oocyte and 4cells, and only in 5-8cells is uptake reduced in KO. Could this result indicate that Slc6a19 functions predominantly in 5-8 cells and ATT other than Slc6a19 functions predominantly in oocytes and 4 cells? The authors appear to be interpreting the results the other way around. The reasons for this interpretation should be described in more detail. 

Q4. At line 246 on page 8, “The rate of uptake of L-[3H]-Pro by oocytes, 4-cell and 5-8-cell stage embryos from WT and Slc6a19-/- mice was examined in the absence and presence of excess unlabelled Pro.”

Please describe more details on how these embryos were generated; does “WT” mean WT female x WT male embryos? and does “Slc6a19-/-“ mean Slc6a19-/- female x Slc6a19-/- male embryos?

Reviewer 2 Report

This manuscript is focused on the function of B0AT1, a neutral amino acid transporter of Pro, Ala and Leu during the preimplantation development by using B0AT1 KO mice. The results showed that Pro uptake by B0AT1 was stage specific, and Pro improved the development ability to blastocyst. These data confused the readers because these data are not sufficient to understand the function of B0AT1 for Pro uptake and embryogenesis. The findings in this study are also not novel and not highly original.

    Therefore, this manuscript is not suitable for publication in Cells in the present form.

Major comments

1)    In Figure 4, the authors stated that there were one or more AATs in WT embryos because excess unlabeled Pro decreased the rate of labeled Pro. Is that right? The ratio of Pro (unlabeled Pro : labeled Pro) was not effect on this rate?

The authors showed that there was no significant difference between No AA and Pro in oocyte and 4 cell of B0AT1 KO embryos, but I think that the number of embryos tested was too small and the variation was too large to be significant difference. The authors should increase the number of embryos examined.

2)     As in Figure 4, the number of embryos tested was too small in terms of the number of embryos tested in Figure 5. Especially, n=3 is too small.

3)     P8 L267 and P8 L274 are contradictory. The authors state at P8 L267 that B0AT1 is a transporter for Pro, Leu and Ala, whereas state at P8 L274 that B0AT1 is a transporter for Pro in 5-8 cells. This contradictory statement should be avoided, as it confuses the readers.

4)     P8 L280-P9 L282: The effects of Ala and Bet in Slc6a19-/- mice should be confirmed. They are very important data to clarify the role of Slc6a19 at 4-cell embryos.

5)     Figure 6: These data are very difficult to understand. In general, embryo developmental rate is expressed as the percentage of embryos that reach each stage relative to the total embryos examined. Therefore, the developmental rate should decrease as development progresses at 2-, 4- and 8-cell, morula, and blastocyst. However, the results in Figure 6 showed a decrease in the cavitation rate on day4, but an increase in the blastocyst rate thereafter. I don't understand why the focus is on cavitation, when you say that in Slc6a19-/- embryos the cavitation rate is lower and cannot be rescued even with the addition of Pro, but the incidence of Bl is the same as WT. Moreover, if the authors want to show that cavitation is inhibited in Slc6a19-/- embryos, should add photos.

6)     P11 L345-346: Why are PROT, GlyT1, PAT1 and PAT2 inactive in Slc6a19-/- oocytes? The data should be provided as evidence. The authors should show the gene expression analysis of those genes in Slc6a19-/- oocyte.

7)     P11 L355-358: Similarly, why are PROT, GlyT1, PAT1 and PAT2 inactive in Slc6a19-/- 4-8cell embryos? The authors should show the gene expression analysis of those genes in Slc6a19-/- 4-8cell embryos.

Mainor comments

1)     P1 L17: Remove the reference in Abstract [1].

2)     P3 L129: Why 120 h is day6? What is 0 h?

3)     P8L249-250: Why the authors determined the presence of one or more AAT(s) for Pro in WT embryos? The results means that the labeled Pro uptake decreased because the unlabeled Pro uptake via B0AT1? Is that right?

4)     P9 L300-302: The authors stated that addition of 0.4 mM Pro to the culture medium largely rescued the decreased development of Slc6a19-/- embryos except at cavitation. That means Pro rescued the development to >5C at day3 and to blastocyst at days 5 and 6. However, judging from the data in Figure 6, the percentages of blastocyst were not different between Slc6a19-/- embryos cultured with or without Pro, that were about 70%. On the other hand, the percentages of blastocyst were lower in WT embryos cultured in Pro than in no AA (90% vs 80%). These results indicated that Pro reduced the developmental ability to blastocyst, not rescued it.

Round 2

Reviewer 1 Report

The authors responded appropriately to the points I raised.

Author Response

Thank you

Reviewer 2 Report

 I still have some unclear matters on the manuscript. Some of these comments can be found below.

Major points

1)    In Figure 4 and P8 L244-248: If B0AT1 is Active, would excess labeled Pro decrease the rate of L-[3H]-Pro uptake or not change it? I think it decrease. Please clear that. Otherwise, I don't understand what you mean by "one or more AATs for Pro were shown to be present (P8 L247-248)".

2)    P11 L329-331: The authors discussed that “Pro uptake in the knockout is not transporter mediated.”. If that is correct, how do the authors explain that WT uptake is reduced by Pro excess and that the rate of Pro uptake is lower than Slc KO? When Slc is activated, "transport through channels that also conduct AA, such as the volume-sensitive anion channels present in WT oocytes" is inactivated? How do the authors explain?

3)    P11 L339-340: Are these results from No AA? Lau excess seems to increase Pro uptake in KO rather than WT.

4)    In Figure 6 and P11 L360-363: The authors mentioned that B0AT expressed on day4 of embryonic development is important for cavitation. If so, why does delayed cavitation occur in Slc KO mice without B0AT1? What is the mechanism? Please discuss it.

As I pointed out previously, if the authors mentioned that the early development is delayed in Slc KO mice, you should show pictures that show this (e.g., WT cavitates on Day 4, but KO remains Morula).

Author Response

Response to major points:

1) In Figure 4 in WT mice, for both oocytes and 4-8-cell embryos, excess unlabelled Pro does indeed reduce uptake of [3H]-Pro. This is expected but it doesn’t define what transporters are involved. It could be BOAT1 and/or any other Pro transporters.

In the KO mice, however, uptake of [3H]-Pro is not reduced by excess unlabelled Pro at either the oocyte or 4-8 cell stage. This suggests that saturable transport at both these stages is, in fact, normally due to BOAT1.

To clarify this, we have reworded L249-253 to read as follows:

“Whereas in oocytes and 4-8 cell embryos from Slc6a19-/- mice, unlabelled Pro failed to prevent L-[3H]-Pro uptake (Figure 4), suggesting B0AT1 is normally an active AAT at these stages in WT mice. Any uptake now in the Slc6a19-/- mice is via a compensatory mechanism, which is not inhibited by the presence of excess unlabelled Pro (see Discussion).”

This rewording hopefully also helps with the next point.

2) Figure 4 does indeed show that “WT uptake [of L-[3H]-Pro] is reduced by [unlabeled] Pro excess”. This, of course, indicates that transport is taking place via one or more ‘traditional’ transporters (such as BOAT1). However, in the Slc KO mice, uptake of L-[3H]-Pro is still ‘rapid’ – it is the same in oocytes and only a little lower in 4-8 cells compared to WT mice. Some sort of compensatory mechanism of uptake is taking place. Since this uptake of L-[3H]-Pro cannot be significantly inhibited by excess unlabelled Pro, this compensatory mechanism cannot be due to any of the traditional transporters. It must be due to some other mechanism which is not saturable or easily inhibitable by competition. We speculate that it is due to “the volume-sensitive anion channel known to be present in oocytes” and cite references [34-36] as support.

3) Yes these are freshly isolated embryos and labelled Pro uptake is being measured in No AA (figure 4B grey bars). The Methods have been adjusted to reflect this (L135). However, in Figure 5, Leu doesn’t “increase Pro uptake in KO rather than WT”. Instead, this figure shows that (i) Leu inhibits the uptake of L-[3H]-Pro in WT mice, which means that whatever transporter is involved it handles both Pro and Leu and (ii) Leu doesn’t inhibit the uptake of L-[3H]-Pro in the KO mice, which shows that the transporter that handled both these AAs in WT is no longer active in the KO.

4) We speculate that delayed cavitation occurs in the KO mice because of the very dynamic expression of AA transporters across pre-implantation development. In other words, in the presence of exogenously added Pro, the absence of BOAT1 at day 4 in the KO mice reduces cavitation expected at this time. However, as time/development proceeds, other transporters are expressed, which then permit Pro uptake and a ‘catch-up process’ can occur. We now speculate briefly on this in L366-369.

We don’t think that any images will be of value to the paper. We have a rigorous scoring process, which was used for generation of the data in figure 6. We didn’t feel that it was necessary to take images at the time.